# Occurrence and Assessment of Organic Pollutants Residues in the Aquatic Environment of the Coastal Sediments

**Sadeq Abdullah Abdo Alkhadher [1,\*], Suhaimi Suratman [1] and Mohamad Pauzi Zakaria [2]**

[1] Institute of Oceanography and Environment, Universiti Malaysia Terengganu, Kuala Nerus 21030, Malaysia; suratman@umt.my

[2] Institute of Ocean and Earth Sciences (IOES), University of Malaya, Kuala Lumpur 50603, Malaysia; pauzi@um.my

\* Correspondence: sadeq.abdo@umt.edu.my

**Abstract:** The current study aimed to monitor organic pollution on island and coastal environments using linear alkylbenzene (LAB). The aquatic environment is affected by the hazardous chemicals discharged through domestic and industrial waste. The distribution, composition, and sources of LABs in the sediments of Port Dickson coast and Pulau Merambong were identified using gas chromatography–mass spectrometry (GC–MS). Chains ranging from long to short (L/S), $C_{13}/C_{12}$ homologs, and internal to external (I/E) congeners were used to define the degradation rate of LABs and the efficacy of wastewater treatment plants. The results of this study revealed that the concentration of LABs in the sites under investigation varied from 67.4 in Pulau Merambong to 255.8 ng g$^{-1}$dw, in Port Dickson. The LAB homologs had a significant difference and a significant percentage of sampling stations had $C_{13}$-LAB homologs. According to the determined LAB ratios (I/E), which ranged from 1.6 in Pulau Merambong to 4.1 in Port Dickson, treated effluents from primary and secondary inputs are being introduced into the aquatic ecosystem of these areas. The degradation of LABs was up to 64% in the interrogated locations. The conclusion is that the wastewater treatment system needs to be improved, and that LAB molecular markers are highly effective in tracing anthropogenic sewage contamination.

**Keywords:** sediment; I/E ratio; wastewater pollution; degradation; molecular marker

## 1. Introduction

The world's ecosystem has faced several dangers and threats over the last decade, and some of it has been lost as a result of mounting pressure from unchecked human exploitation of natural resources [1]. An essential component of these hazards, stemming from the widespread practice of surfactants in daily life, industry, and agriculture during the past few years, is the pollution of coastal and riverine ecosystems. In spite of Malaysia's growing urbanisation and industry, the country's sanitary sewer systems, even those in populated and developed areas, are of low quality [2,3]. Several rivers and estuary areas acquire land-based contaminants, such as industrial and municipal sewage. The monitoring of these ecosystems is important to identify the occurrence of possibly hazardous pollutants and their adverse influences on the coastal environment, hence the need to provide the necessary data for possible environmental management and preservation [4].

Molecular markers are persistent organic molecules with source-specific structures or isotopic compositions that serve as carriers of source information for specific organic materials or environmental contamination [5]. These substances can have either natural or man-made sources, and each source's characteristics are particular to the molecular markers it produces. Additionally, because molecular markers have signatures from numerous previous geological eras and natural processes, they can be used to identify the origins of organic compounds [6].

The final criteria for selecting a marker are determined by the marker's specific source of influence, its accessibility in the environment, and its persistence in the environment even after undergoing environmental changes [7]. Recently, there has been a lot of interest in the use of genetic markers to determine the origins of sewage contamination [8]. Worldwide sources of sewage contamination have been effectively studied using molecular markers such as faecal sterols and linear alkylbenzenes (LABs) instead of microbiological approaches [9]. In sediments, faecal sterols and LABs tend to adhere to particles and build up [10,11]. Both of these substances are hydrophobic and persistent. Coprostanol is produced in the intestine by the hydrogenation of cholesterol and is a key indicator of human faecal contamination. There is a positive relationship between coprostanol concentrations and the number of faecal bacteria [12]. For example, a substantial link was found between coprostanol levels and *E. coli* abundance in the environments of Vietnam and Malaysia [8]. When coprostanol is injected into an aquatic environment, it absorbs particulate matter and settles on the soil. Because it persists in sediment, coprostanol is a faecal contamination indicator.

Domestic pollution can be measured using organic chemical compounds such as linear alkylbenzenes (LABs), with a formula of $C_6H_5$-$C_nH_{2n}$ + l, n = 10–14, which are the key ingredients in detergents (linear alkylbenzene sulfonates; LASs) [13]. In the 1960s, branching alkylbenzene chemicals had totally converted to LABs for detergent manufacture because of their superior biodegradability and affordability. Due to insufficient sulfonation, LABs are released into the marine environment in a large amount of untreated residential and industrial waste [14,15] Therefore, LABs are utilised as anthropogenic pollution indicators because of their enduring and high affinities to household and industrial effluents [3,16]. As a result of the unique phenyl substitution to the straight alkyl chain in their structured exterior and inner isomers, LABs are also employed to show the degree of deterioration in the sediments as well as particulate matters [17]. Furthermore, LABs are utilised to identify the length of stay and the sorts of household and industry wastewater, such as primary sewage and secondary effluents, discharged into an aquatic ecosystem [10,18].

Malaysia has experienced tremendous economic and demographic expansion over the past three decades, while simultaneously developing rapidly in terms of industrialisation, urbanisation, and motorisation. As a result, there are now more dangers and potential negative impacts that could damage this nation, particularly as a result of the toxins linked to urbanisation and industrialisation that are gradually released into the aquatic ecosystem [13].

In Malaysia, the sources and degree of sewage or wastewater contamination vary from one area to another [19]. Due to the abundance of population growth and waste treatment plants on Malaysia's western coast, where the investigated areas are located, residential waste is increasingly present over point sources such as factories, in addition to non-point sources such as discharge [20]. Higher health risks to people and the ecosystem would result from increased inputs of untreated wastewater into river and coastal ecosystems, particularly in slum regions and locations that are overcrowded, constrained, or without wastewater treatment systems. Because of the rising degree of contamination in the rivers, which, in turn, causes severe pollution in Peninsular Malaysia's coastline, represented in this study by Port Dickson and Pulau Merambong, the marine ecosystem is exposed to the accumulation of organic pollutants, including LABs. Studies on the distribution and transportation routes of LABs in Malaysia's sedimentary environment are, however, scarce.

Researchers routinely analyse various environmental samples, including sediments, to better understand the levels of LAB pollution in the riverine environments. The current study's goals are to evaluate anthropogenic contributions in the selected areas using LABs as molecular markers. The researched location underwent the measurement of LAB distribution, concentration, and levels of degradation. Additionally, efforts were made to improve the efficiency of the existing sewage treatment plants (STPs).

## 2. Methods and Materials

### 2.1. Sampling

The studied sites are located in Peninsular Malaysia, including coasts and islands (Figure 1).

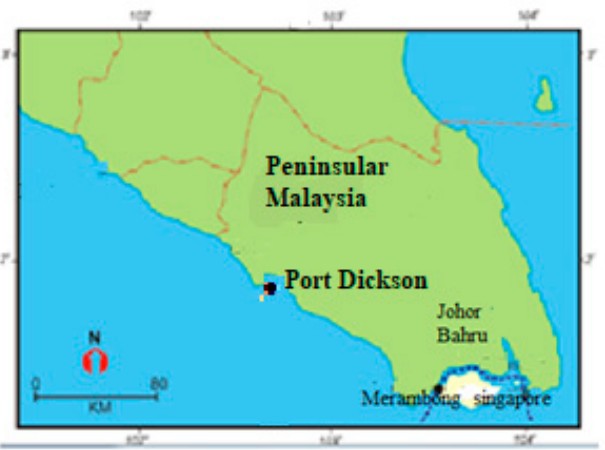

(**a**)

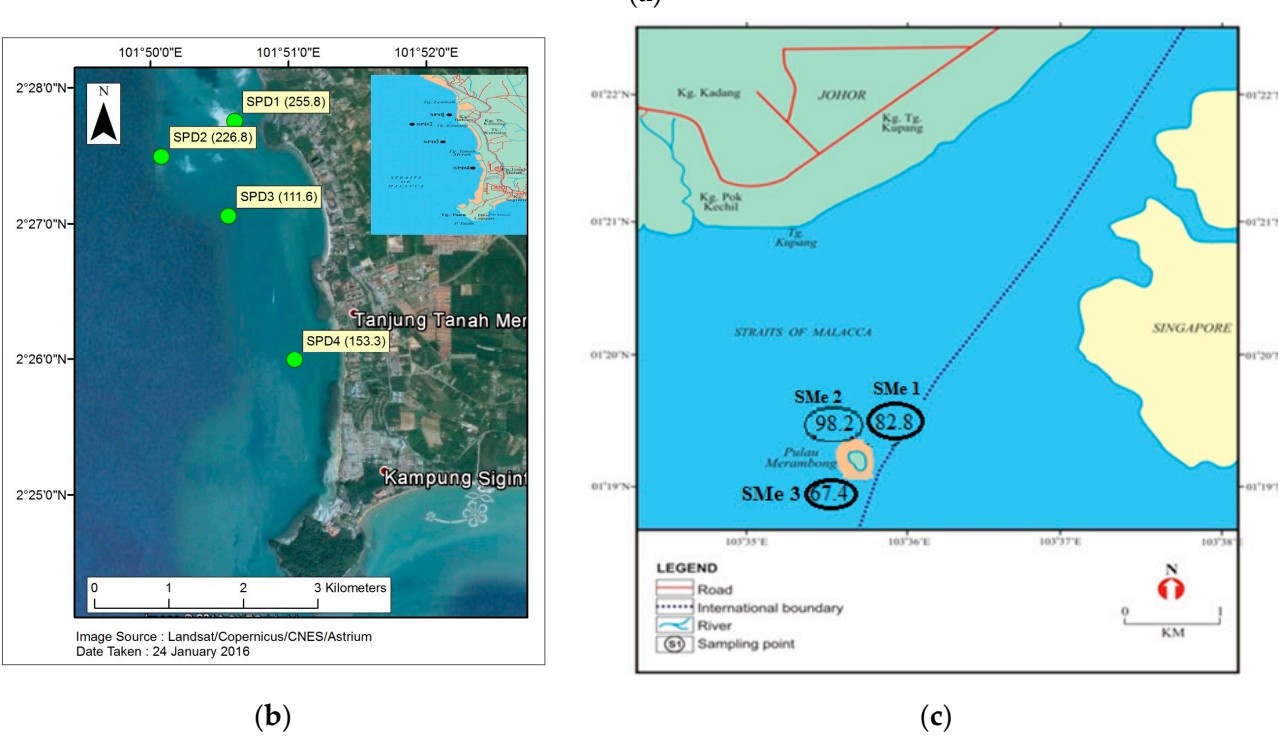

(**b**)                                                                 (**c**)

**Figure 1.** General site of the study area, showing (**a**) a map of the Peninsular Malaysia, (**b**) the port Dickson coast and (**c**) the Pulau Merambong.

The description of the study locations of the Port Dickson coast with station codes (SPD1, SPD2, SPD3, SPD3, SPD4) and Pulau Merambong with station codes (SMe1, SMe2, SMe3) are illustrated in Table 1.

**Table 1.** Sampling sites along the Port Dickson coast and Pulau Merambong.

| Sample Name | Geographical Coordination | Location Type | Weather Condition | Site Description |
|---|---|---|---|---|
| SPD1 | 2°27′882″ N 101°50.687″ E | Beach | Cloudy | Urban and tourism area |
| SPD2 | 2°27′638″ N 101°50.7′35″ E | Beach | Cloudy | Urban and tourism area |
| SPD3 | 2° 28′047″ N 101°50.7′96″ E | Beach | Cloudy | Urban and tourism area |
| SPD4 | 2°27′943″ N 101°50.8′10″ E | Beach | Cloudy | Urban and tourism area |
| SMe1 | 1°22′76.0″ N 103°38′08.8″ E | Island | Sunny | Uninhabited area |
| SMe2 | 1°20′61.7″ N 103°36′72.0″ E | Island | Sunny | Uninhabited area |
| SMe3 | 1°20′40.3″ N 103°36′27.7″ E | Island | Sunny | Uninhabited area |

The studied areas were chosen based on the occurrence of low and high levels of anthropogenic activities alongside the selected areas. The collection of the surface sediments was conducted to evaluate the most recent anthropogenic LAB inputs in the aforementioned regions. Each sediment sample's top 4 cm were obtained using an Ekman dredge sampler. Samples of sediments were then put inside the stainless-steel containers that had already been pre-solvent washed, put in a cooler box, taken to the lab, and kept there in a freezer at $-20\ ^\circ$C.

### 2.2. Chemical Analysis

Two columns were applied for extraction of hydrocarbons from the sediments. The first column was used for purification to remove polar mixtures, followed by the second one that was utilised for fractionation to obtain the anticipated extract, according to descriptions in the literature [2,21].

In total, 250 mL of dichloromethane (DCM) was poured into a cellulose thimble containing 10 g of dried sediment for 10 h in a soxhlet apparatus [22]. Prior to extraction, each sample had a fixed volume of 50 μL of "1-Cn" LAB as surrogate standards (SS) for recovery correction of LAB individuals. Before being moved to the top of a chromatography column of 0.9 cm i.d, 9 cm height filled with 5% $H_2O$-deactivated silica gel (60–200 mesh size, sigma chemical company, USA), the extract was reduced in volume with a rotary evaporator. In total, 20 mL of a pure 3:1 *v/v* hexane/DCM mixture were used to elute the extracted hydrocarbons in the first step, and they were then reduced to 2 mL. In the second step, on a silica gel column that had been fully activated (0.47 cm i.d., 18 cm height), 4 mL of hexane were used to get the LAB fraction. The reduced extract was then moved to a 2 mL amber vial and further reduced using a gentle stream of nitrogen until dry. In LAB fraction, the internal standards (IS = biphenyl-d10, $m/z$ = 164) were added before the gas chromatography–mass spectrometry (GC-MS) measurement.

An Agilent Technologies 7890A series gas chromatograph was linked to a 5975 MSD split/splitless injector in order to identify LAB individuals. Briefly, a 30-m fused silica capillary column with a 0.25-mm (i.d.) was used in conjunction with a DB-5MS capillary column with a 0.25-m film thickness. Helium served as a carrier gas and was maintained at a pressure of 60 kg cm$^2$ at all times. Chromatographs of LABs were detected at $m/z$ = 91, 92, and 105, and the mass spectrum data were collected using the selective ion monitoring (SIM) mode. The ionization process in the GC–MS was conducted with a setting of 70 eV and employed an external source at 200 °C and an electron multiplier voltage of approximately 1250 eV. Following a 1-min purge, the sample injection in splitless mode was carried out with an injection port kept at 300 °C.

### 2.3. Quality Assurance and Control

In the allowed range (between 60 and 120%), surrogate standards for LABs (1-Cn LABs) were recovered with a reasonable efficiency percentage with only a minor loss of the target compounds occurring during the analytical procedure of the LABs. The range of recoveries for LAB surrogates for all sediment in this analysis was between 87 and 98%. To further avoid any possible sources of cross pollution with the analysis steps, there was a blank sample (four samples per batch) that contained all the standards including

surrogate standards, internal standards, and native standards (SS, IS, and NS) that were present in a regular sample. Every day, freshly manufactured SS, IS, and NS were added to sediment samples at known concentrations. To find the target LAB congeners, at $m/z = 91$, 92, and 105, a GC–MS was employed in SIM. A 5-point calibration curve resulting from five concentrations of LAB standard mixture (SS, IS, NS) between 0.25 and 5.0 ppm was used to identify the LAB target congeners. The limits of quantitation (LOQ) and detection (LOD) were obtained via dividing the lowest concentrations of each calibration curve [23]. They fell between 0.1–2 and 0.02–0.1 ng g−1, respectively.

### 2.4. Statistical Analysis

The statistical analysis was conducted using IBM®SPSS-25 software, employing analysis of variance (ANOVA) to illustrate the significance of variations in LAB distribution between locations at a threshold of $p < 0.05$. The Pearson correlation coefficient was used to correlate the sampling locations. Standardised regression coefficients were employed to determine the possible contribution of various LAB sources.

### 2.5. Total Organic Carbon Method (TOC)

The samples were crushed in a mortar and pestle after being temporarily dried in a 60 °C oven for the entire night. In order to remove the inorganic carbon, each weighted dried sediment sample (1–2 g) was completely moistened with 1M HCl. The samples were dried for 10 h at 100 °C to remove any remaining HCl. The TOC in the sediments was analyzed using a LECO CR-412 analyzer, as described by [24]. Table 2 has the calculated TOC%.

**Table 2.** LABs concentration (ng g$^{-1}$ dw) and relative compound ratios in the Port Dickson coast and Pulau Merambong.

| Compound | [b] SPD1 | SPD2 | SPD3 | SPD4 | SMe1 | SMe2 | SMe3 |
|---|---|---|---|---|---|---|---|
| [a] $C_{10}$-LABs (ng g$^{-1}$ dw) | 17.2 | 14.9 | 4.2 | 6.2 | 7.2 | 8.8 | 6.4 |
| $C_{11}$-LABs (ng g$^{-1}$ dw) | 66.7 | 56.9 | 19.0 | 30.7 | 13.1 | 14.3 | 11.4 |
| $C_{12}$-LABs (ng g$^{-1}$ dw) | 68.6 | 61.0 | 24.0 | 38.0 | 16.4 | 19.4 | 14.1 |
| $C_{13}$-LABs (ng g$^{-1}$ dw) | 86.0 | 78.4 | 51.2 | 63.6 | 23.9 | 31.9 | 18.4 |
| $C_{14}$-LABs (ng g$^{-1}$ dw) | 17.4 | 15.6 | 13.2 | 14.8 | 22.2 | 23.8 | 17.1 |
| LABs (ng g$^{-1}$ dw) | 255.8 | 226.8 | 111.6 | 153.3 | 82.8 | 98.2 | 67.4 |
| [c] I/E | 4.1 | 3.7 | 2.0 | 2.6 | 1.7 | 1.7 | 1.6 |
| [d] L/S | 1.6 | 1.6 | 1.7 | 1.6 | 2.0 | 2.0 | 2.0 |
| [e] $C_{13}/C_{12}$ | 9.7 | 9.5 | 13.7 | 14.3 | 4.9 | 5.7 | 4.1 |
| [f] LAB Degradation (%) | 64 | 61 | 40 | 48 | 34 | 34 | 33 |
| [g] TOC (mg/g) | 19.7 | 22.7 | 14.7 | 18.9 | 25.4 | 6.5 | 35.1 |

[a] $C_{10}$-LAB. Sum of the 26LAB congeners ranging from 5-$C_{10}$ to 2-$C_{10}$. [b] SJB1, the first letter indicates the station, the second and third letters represent the first letters of location name, the numbers 1,2,3,4, indicate the first, second, third and fourth station. for each location respectively. [c] I/E ($C_{12}$-LABs), ratio of (6-$C_{12}$LAB + 5-$C_{12}$LAB) relative to (4-$C_{12}$LAB + 3-$C_{12}$LAB + 2-$C_{12}$LAB). [d] L/S, ratio of (5-$C_{13}$LAB + 5-$C_{12}$LAB) relative to (5-$C_{11}$LAB + 5-$C_{10}$LAB). [e] $C_{13}/C_{12}$, ratio of (6-,5-,4-,3- and 2-$C_{13}$/(6-, 5-, 4-, 3-, and 2-$C_{12}$LAB). [f] LAB Degradation (%), LAB deg = $81 \times \log$(I/E ratio) +15 ($r^2$ = 0.96). [g] TOC, Total organic carbon amount.

## 3. Result and Discussion

### 3.1. Composition, Distribution and Concentration

The 26 different isomers of the LABs have been written as n-Cm LAB, where "n" is the carbon number and "m" is the location of the alkyl chain phenyl. The total number of LABs $C_{10}$–$C_{14}$ found in samples from all the sites under study are reported in Figure 2.

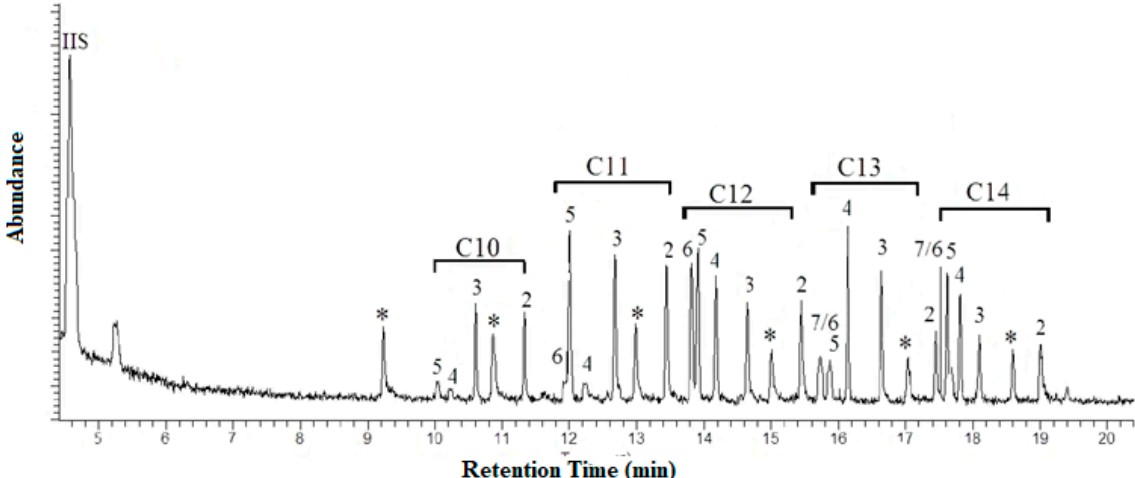

**Figure 2.** Gas chromatograms of LABs in surface sediments of the Port Dickson coast and the Pulau Merambong. IIS (Internal Injection Standard-biphenyl, d10), Surrogates 1-Cn-LABs (n: 8–14 from left to right) indicated by asterisks. Numbers on the peaks indicate the phenyl substituted position on the alkyl chain.

The composition profiles of the LAB homologs at the sample sites are shown in Figure 3.

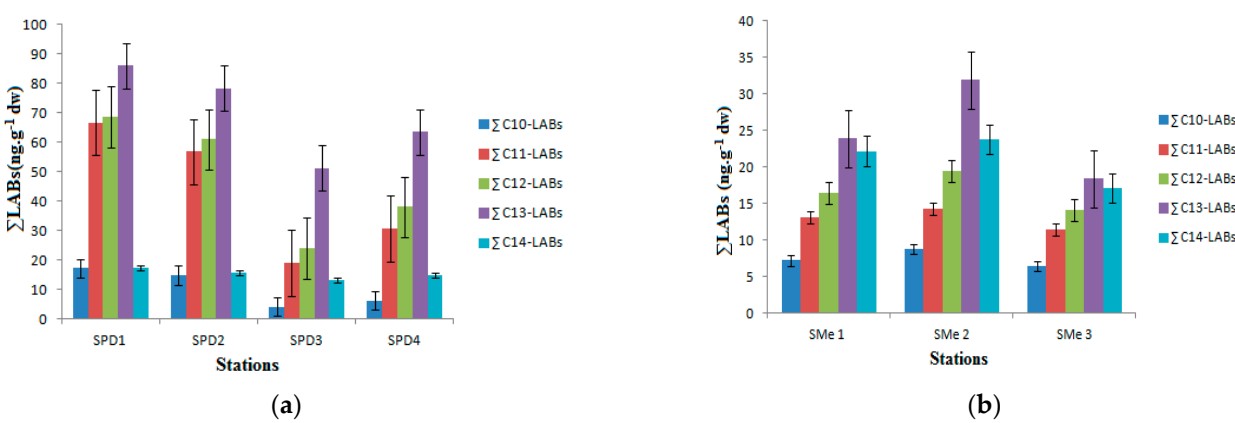

(**a**)                                                                 (**b**)

**Figure 3.** Compositional profile of LABs in (**a**) Compositional profiles of LABs in Port Dickson. (**b**) Compositional profiles in Merambong.

High levels of $C_{13}$ homologs were found in all study stations. In most places, the concentrations of longer chain LABs (LC- homologs), such as $C_{13}$ and $C_{14}$, were higher than those of short chain homologs (SC- homologs), such as $C_{10}$ and $C_{11}$. The abundance of lengthy chain LABs ($C_{13}$ and $C_{14}$), compared with short LAB homologs ($C_{10}$ and $C_{11}$), tended to prevail in marine environments, demonstrating that LABs have a long lateral transport in this area. When the composition of the LABs was carefully analysed, it was found that the first station along the Port Dickson coast (SPD1) had a significant concentration of $C_{13}$ homologs, indicating that these chemicals undergo anaerobic breakdown [3,25]. This chain length distribution is in line with earlier research published by [26,27], which discovered that surface sediments included an increase in the number of LABs with longer alkyl chains ($C_{13}$-$C_{14}$) compared with sewage effluents because long chains are less volatile and have lower vapour pressure than those with shorter chains. With increased chain length, the compositional distribution of LABs returned to a high hydrophobicity, as seen by the low abundance of $C_{10}$ and $C_{11}$ homologs (short-chain LABs) in all researched sites [28]. The distribution of LAB homologs in the sediments of these research sites demonstrated lower $C_{10}$ homologs compared with the detergents researched by [29].

As shown in Table 2, the presence of LABs $C_{10\text{-}14}$ in the surface sediments varied between 67.4 in Pulau Merambong's SMe3 and 255.8 ng g$^{-1}$ dw in Port Dickson's SPD1 (Figure 4), demonstrating that SMe3 in Pulau Merambong had the lowest concentration of LABs, whereas the first station (SPD1) of Port Dickson had the highest.

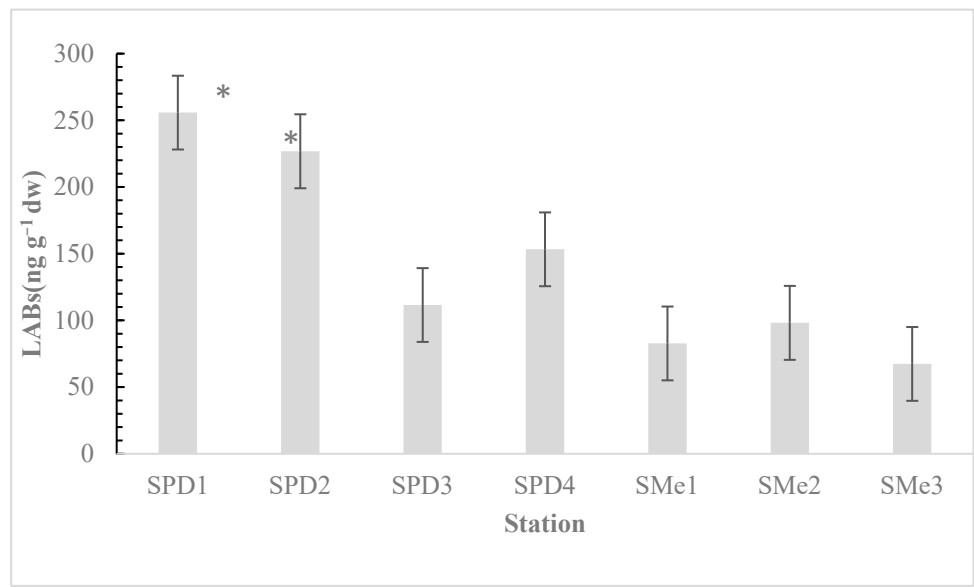

**Figure 4.** Concentration of LABs in the Port Dickson coast and Merambong Samples, Standard error bars are shown. * Concentration is significant at the 0.05 level.

The study's findings show a significant Pearson correlation between the concentration of LABs ($r = 0.75$) among the locations under investigation, indicating that LABs are able to be used to indicate anthropogenic contamination in the maritime environment (Table 3).

**Table 3.** Pearson correlation coefficients between total LABs concentration in the Port Dickson coast and Pulau Merambong. Significant Pearson correlation ($p < 0.05$).

|  | Location | Total LABs Concentration |
|---|---|---|
| Pearson Correlation | 1 | 0.75 ** |
| Sig. (2-tailed) |  | 0.05 |
| N | 7 | 7 |

** Correlation is significant at the 0.05 level (2-tailed).

At $p < 0.05$, there was consistently a significant difference in LAB concentrations between the research sites (Table 4).

**Table 4.** Overall analysis of variance for surface sediments of Port Dickson coast and Pulau Merambong.

| Source | DF | Sum of Square | Mean Square | F Value | Sig. * |
|---|---|---|---|---|---|
| Side | 1 | 92,278.39 | 9227.39 | 10.11 | <0.05 |
| Locations | 2 | 114,597.32 | 2291.46 | 50.20 | <0.05 |
| Error | 25 | 0.00 | 0.00 |  |  |
| Corrected Total | 7 | 143,098.47 |  |  |  |

* Correlation is significant at the 0.05 level (1-tailed).

The results of this investigation showed that the Port Dickson's LAB levels are greater than those in Pulau Merambong. These regional distributions are most likely the result of increased industrialisation and urbanisation near the sampling sites. The LAB distribution in the investigated sediments followed the pattern: Port Dickson > Pulau Merambong, demonstrating that the geographic site does affect LAB dispersion.

The investigated sites showed lower LAB concentrations compared with those reported in previous studies on the Sarawak River, Sembulan River, and Anzali Wetland [19,30], but similar LAB concentrations were reported in the Kim Kim estuary and Southern Brazil [11,31]. Compared with levels of LABs in the sediments from other locations in Malaysia and throughout the world, the concentrations of LABs in the examined areas under study were generally low to moderate (Table 5).

**Table 5.** Total concentrations of LABs from different areas around Malaysia and the world.

| Location | N | Maximum LABs (ng/g) [a] | I/E Ratio [b] | Degradation [c] (%) | Reference |
|---|---|---|---|---|---|
| South Atlantic Estuary | 15 | 210 | 2.5 | 47 | [32] |
| Southern Brazil | 3 | 15.3 | 1.4 | 27 | [31] |
| Humber Estuary and Wash, UK | 18 | 84.8 | 2.1 | 41 | [33] |
| Anzali Wetland, Iran | 167 | 109,000 | 1.3 | 24 | [30] |
| Malacca, Malaysia | 1 | 1080 | 2.0 | 39 | [11] |
| Muar River, Malaysia | 1 | 32 | 2.8 | 51 | [11] |
| Penang Estuary, Malaysia | 1 | 3000 | 1.5 | 29 | [11] |
| Prai River, Malaysia | 1 | 25 | 3.4 | 58 | [11] |
| Kim Kim River, Malaysia | 1 | 122 | 1.8 | 36 | [11] |
| Kim Kim Estuary, Malaysia | 1 | 6 | 1.2 | 21 | [11] |
| Nibong Tebal, Malaysia | 1 | 168 | 2.1 | 41 | [11] |
| Indonesia | 20 | 42,600 | 2.1 | 41 | [11] |
| Sarawak River, Malaysia | 9 | 7390 | 1.0 | 15 | [19] |
| SembulanRiver, Malaysia | 6 | 5570 | 1.8 | 36 | [19] |
| Zhujiang River | 11 | 2330 | 1.5 | 29 | [29] |
| Dongjiang River | 10 | 566 | 1.9 | 38 | [29] |
| Xijiang River | 8 | 69.4 | 1 | 15 | [29] |
| Pearl River Estuary | 8 | 26 | 1.5 | 29 | [29] |
| South China Sea | 28 | 23 | 0.9 | 11 | [29] |
| The Pearl River Delta | 96 | 11,200 | 1.7 | 34 | [34] |
| Santos Bay, Brazil | 14 | 117 | 2.9 | 55 | [35] |
| Dongjiang River | 45 | 410 | 1.4 | 27 | [36] |
| Outfalls of paper mills | 3 | 3270 | 1.3 | 24 | [36] |
| Jakarta Bay | 7 | 86,800 | 0.9 | 12 | [37] |
| Tokyo Bay | 2 | 1110 | 2.8 | 51 | [37] |
| Detergents | 10 | 5,300,000 | 1.7 | 34 | [33] |

[a] LAB = sum of concentrations of all secondary LAB congeners having $C_{10}$–$C_{14}$ alkyl chain. [b] I/E = $(6\_C_{12} + 5\_C_{12})/4\_C_{12} + 3\_C_{12} + 2\_C_{12})$. [c] LAB deg = $81 \times \log$ (I/E ratio) + 15 ($r^2 = 0.96$). N—The number of samples.

The most likely explanation for their spatial distribution in these sites was found to be the discharge of wastewater effluents and their dilution with organic particle materials that may change LAB distribution in coastal environments [32,38]. However, the spatial distribution of LABs in some places could be due to factors such as a high rate of urbanisation, industry, and inadequate sewage systems [20,39].

### 3.2. TOC Evaluation

If introduced to an aquatic environment, LABs are expected to attach firmly to organic molecules due to their high hydrophobicity. As a result, TOC in the sediments was associated with the level of LAB [40].

All sediment samples were TOC assessed, and most stations in Port Dickson and Pulau Merambong showed a relationship between LABs and TOC ($R^2 = 0.64$, 0–0.97; Figure 5), respectively.

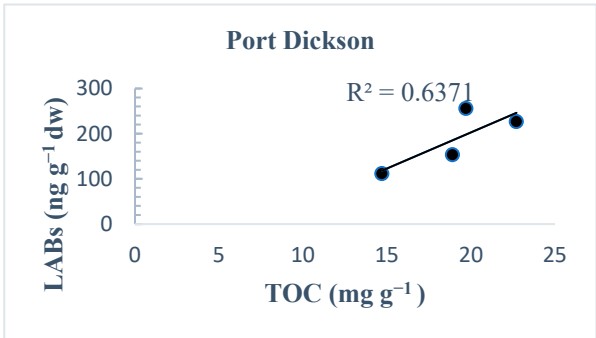
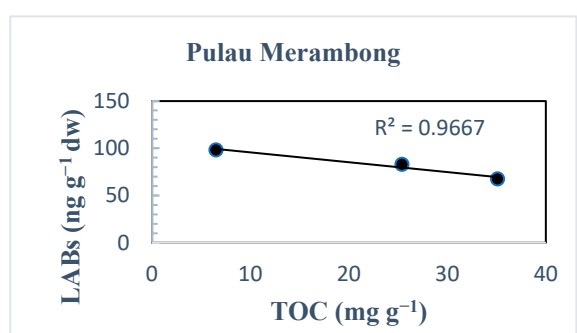

**Figure 5.** Scatter plots of LABs and TOC in sediment samples of the Port Dickson coast and Merambong.

These findings suggest that TOC may play a crucial role in regulating the spatial distribution of LABs from the land to the sampling stations. This is in line with the Dongjiang River TOC results, which showed a linear correlation between LAB concentrations and TOC content ($R^2 = 0.82$) and revealed that wastewater discharge was the primary source of organic carbon in the sediments. Prior studies have demonstrated a significant correlation between TOC and the quantities of hydrophobic organic compounds [41–43].

The Pulau Merambong's SMe2 had TOC with 6.5 mg g$^{-1}$, and SMe3 in the same island had TOC with 35.1 mg g$^{-1}$, implying that SMe2 of Pulau Merambong had the lowest concentration of TOC, whereas SMe3 on the same island had the highest. The TOC results are distinct from those in the Selangor and Perak rivers in West Malaysia with $R^2 = 0.008$ and 0.17, respectively [2,22]. This shows that the intensification of anthropogenic input in those locations was a contributing factor to the distribution of LABs in the Selangor and Perak rivers, also demonstrating that the TOC was a determining factor for the distribution of LABs in this study.

### 3.3. Assessment of LAB Degradation and Effluent Treatment Efficiency

The ratios of LABs affect the sources of LABs and the degradation of aquatic ecosystems. The identification of LAB sources in aquatic environments can be undertaken with the help of molecular ratios and LAB ratios have been used by researchers to identify sources in sediments in the literature [18,22].

One of these ratios is an internal (I) isomer to the external (E) one (I/E ratio), since the internal isomer of LABs is more difficult to degrade than the external one [11]. The I/E ratios increase during LAB biodegradation under aerobic conditions [18]. As a result, the use of LAB ratios as measurements of the degree of LAB degradation has been suggested. I/E ratios are used to gauge how well wastewater is treated and how quickly LABs degrade in aquatic environments [18,44].

The I/E ratios indicate that primary and secondary effluents are being released into the aquatic environments of the study area. In our study, LAB degradation was found in low levels in the surface sediments at SMe3 in Pulau Merambong while high levels were found at SPD1 and SPD2 in Port Dickson (see Table 2). These results showed that Port Dickson had the highest LAB biodegradation at 64%, while Pulau Merambong had the lowest at 33%, suggesting that Port Dickson has a higher level of selective biodegradation of LAB species than Pulau Merambong, which has weak degradation and inadequately treated wastewater in some stations.

The I/E ratios ranged from 1.6 in SMe3 of Pulau Merambong to 4.1 in SPD1 of the Port Dickson coast, indicating that the secondary treatment processes were the main sources of LABs in Port Dickson, while Pulau Merambong provided the primary-treated sewage feed for the majority of the stations. This is explained by the fact that Port Dickson has more efficient STPs (Figure 6).

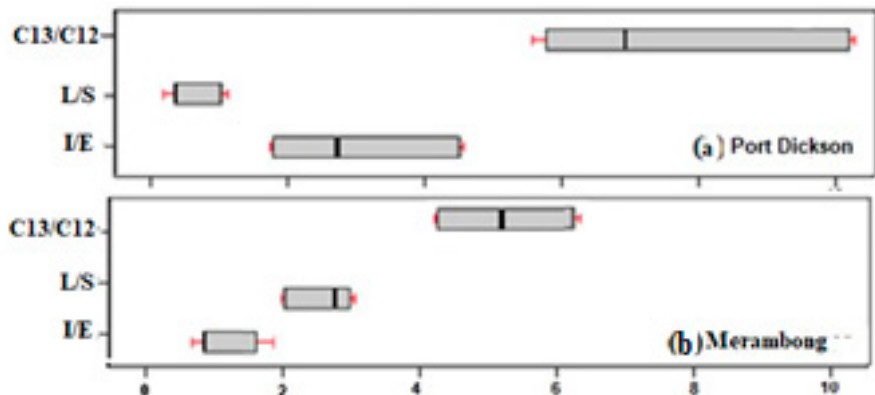

**Figure 6.** Values of the $C_{13}/C_{12}$, L/S, and I/E ratios in the samples of the Port Dickson coast and Pulau Merambong.

LABs can be released from a variety of sources, such as residential or industrial waste. These wastewaters have a strong affinity to particulate matter, such as sewage particles and organic matter, and may therefore improve the LASs' transport from their initial sources to their later transport into the aquatic ecosystem [36,45,46].

LAB biodegradation in the investigated areas was assessed further using the other two ratios (L/S, $C_{13}/C_{12}$) that have been proposed to represent the level of sewage treatment and the degree of LAB degradation [29]. Pulau Merambong had greater L/S and $C_{13}/C_{12}$ levels than the Port Dickson stations (Figure 6). Our results showed that the L/S ratio in the sediment samples from the studied sites varied from 1.6 at the first, second, and fourth stations of Port Dickson (SPD1, SPD2, SPD4) to 2.0 at all stations of Pulau Merambong. This ratio was higher than the value of 1.8 reported in detergents [34], which suggests an enhancement in the degradation of LABs in these areas. The $C_{13}/C_{12}$ ratio was also greater than the ratio of 1.7 observed in the study by [47], ranging from 4.1 at the third station (SMe3) in Pulau Merambong to 14.3 at the fourth station in Port Dickson (SPD4). No significant differences were observed between the I/E and L/S ratios based on the quantitative analysis using Duncan grouping (Table 6).

**Table 6.** Quantitative analysis of LABs ratios of Port Dickson coast and Pulau Merambong.

| Duncan Grouping | Mean | N | Ratio |
|---|---|---|---|
| A | 5.2434 | 7 | $C_{13}/C_{12}$ |
| B | 1.9178 | 7 | I/E |
| B | 1.7363 | 7 | L/S |

However, there was a difference observed between the $C_{13}/C_{12}$ ratios and I/E or L/S ratios, which could be attributed to the differences in ratio sensitivity.

Due to the increased use of the shoreline areas for fishing activity by locals in recent years, there has been a significant direct discharge from ferries and vessels [48]. LABs increase in coastal and island sediments due to detergent waste discharge, and washing from ferries and boats can affect molecular indices to some extent. The flushing of urban materials into estuaries and the marine environment through drains and coasts is exacerbated by heavy rainfall and flash floods. As a result, the sedimentary habitats of Pulau Merambomg were found to have a high signature of primary sources.

The LAB ratios demonstrated how quickly LABs are degrading in the studied areas.

Compared with the markers found in the Selangor River sediments by [22], this study found higher levels of I/E (4.1 vs. 0.6), L/S (4.1 vs. 2.3), and $C_{13}/C_{12}$ (14.3 vs. 2.1), with a particularly greater concentration of $C_{13}$-LABs, which demonstrated that sedimentary LABs had a high rate of biodegradation.

Thus, L/S and $C_{13}/C_{12}$ ratios are suggested as more sensitive indicators of biodegradation and the effectiveness of wastewater treatment in the sediments of the research area. However, the evaluation of biodegradation can be influenced by environmental conditions and the initial concentration of LABs. Fluvial transport is the primary source of the release of terrestrial LABs into the aquatic environment. Moreover, this is in line with earlier published anthropogenic PAHs [7]. The wastewater from places near the sampling stations may mix with organic materials before it reaches the test site since it is dumped into the bay waters.

## 4. Conclusions

A significant difference ($p < 0.05$) in LAB levels was found between the Port Dickson coast and Pulau Merambong stations, with the concentration ranging from 67.4 ng·g$^{-1}$dw in Pulau Merambong's to 255.8 ng·g$^{-1}$dw in Port Dickson's sediments. In comparison with other sampling locations, the first station along the Port Dickson coast reported the greatest number of LABs, whereas the third station of Pulau Merambong had the lowest. The SC- homologs, which are less common than the LC- ones, were found to have the lowest concentration at the third station of Pulau Merambong. I/E ratio data revealed that LABs originated from a variety of primary to secondary sewage treatment types in Pulau Merambong and Port Dickson, respectively. The lowest I/E ratio of 1.6 was observed in Pulau Merambong, which may be attributed to the influence of raw sewage in this region. On the other hand, a high I/E ratio of 4.1 was noted in Port Dickson. The results showed that untreated industrial and residential waste issues may persist for a long time as the population increases due to inadequate STPs resulting from a low I/E ratio. It is expected that there will be an increase in industrial and urban wastewater discharge levels along the islands and coasts of Malaysia in the near future. To address environmental concerns and improve public health in the coastal areas, regular assessments of wastewater pollution and upgrades to sewage systems should be implemented.

**Author Contributions:** Conceptualization, S.A.A.A.; Software, S.S.; Formal analysis, S.A.A.A. and M.P.Z.; Investigation, S.A.A.A.; Resources, S.S.; Data curation, S.S.; Writing—original draft, S.A.A.A.; Writing—review & editing, S.S.; Visualization, M.P.Z.; Project administration, M.P.Z.; Funding acquisition, M.P.Z. All authors have read and agreed to the published version of the manuscript.

**Funding:** The postdoctoral scheme grant from University Malaysia Terengganu (UMT) and the Inisiatif Putra Berkumpulan Grant from the University of Perak (UPM) are funding this study (9412401).

**Institutional Review Board Statement:** Not applicable.

**Informed Consent Statement:** Not applicable.

**Data Availability Statement:** Not applicable.

**Acknowledgments:** The authors would like to acknowledge University Malaysia Terengganu (UMT) and the Inisiatif Putra Berkumpulan Grant from the University of Perak (UPM) are funding this study (9412401).

**Conflicts of Interest:** The authors declare no conflict of interest.

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
