# Peer review of "Occurrence and Assessment of Organic Pollutants Residues in the Aquatic Environment of the Coastal Sediments"

_sustainability, doi:10.3390/su15108365_

Round 1

Reviewer 1 Report

The manuscript evaluates the occurrence and Assessment of Organic Pollutants Residues in the Aquatic Environment of Malesia, in Port Dickson coast and Pulau Merambong which very different level of anthropogenic impact. Even if, the organic chemistry is not my expertise field and some concepts are not easy for me, the article reads well and is well structured, and the results are interesting. However, some revisions are necessary before the publication.

Introduction

“Due to insufficient sulfonation, LABs are released into the marine environment in a large amount of no treated household and industrial wastes. Therefore, LABs are utilised as anthropogenic pollution indicators because of their enduring and high affinities to industrial effluents”

In the first sentence the authors referred both household and industrial wastes, in the second just to industrial effluent. Is it not incoherent?

“Due to less growth on Malaysia's eastern coast, the majority of the country's pollution originates in urban and industrial areas”

Please clarify, in which areas is the greater contamination? Industrial and urban areas, but where?

The researched location underwent measurement of LAB distribution, concentration, and levels of degradation as well as improving the efficiency of the existing STPs.

What are STPs? It is the first time this acronym is reported I the text.

Methods and Materials

QUALITY ASSURANCE AND CONTROL

To further avoid any possible sources of cross pollution with the analysis steps, there was a blank sample (four samples per batch) that contained all the standards (SS, IS, and NS) that were present in a regular sample.

NS was not described.

TOTAL ORGANIC CARBON METHOD (TOC)

In order to completely moisten each weighted dried sediment sample (1-2 g) with 1M HCl, inorganic carbon was removed.

I do not understand, please, rewrite this sentence.

Results and discussion

COMPOSITION, DISTRIBUTION AND CONCENTRATION of LAB

Total number of LABs C10–C14 found in samples from all sites under study (Fig.2).

Maybe “Total number of LABs C10–C14 found in samples from all sites under study were reported in Figure 2”

Lengthy-chains LAB (C13 and C14), as opposed to other LAB homologs (C10 and C11), tend to prevail in marine environments, demonstrating that LABs have long lateral transit in this area.

What the authors exactly mean with “demonstrating that LABs have long lateral transit in this area”. Please, explain better.

Table 2 demonstrates the presence of LABs C10-14 in surface sediments varied between 67.4 in Pulau Merambong's SMe3 to 255.8 ng g−1dw in Port Dickson's SPD1 (Fig.4). SMe3 in Pulau Merambong had the lowest concentration of LABs, whereas first station (SPD1) of Port Dickson had the highest.

These two sentences are redundant, please merge them.

TOC EVALUATION

All sediment samples had TOC assessed, and several locations, including Port Dick-son and Pulau Merambong had a relationship between LABs and TOC

What the authors mean with several locations?

SMe2 of Pulau Merambong had the lowest concentration of TOC, whereas SMe3 in the same island had the highest. The Pulau Merambong's SMe2 had TOC with 6.5 mg g-1 and SMe3 in the same island had TOC with 35.1 mg/g..

Please, merge these two sentences.

ASSESSMENT OF LAB DEGRADATION AND EFFLUENTS TREATMENTS EFFICIENCY

please, try to explain better the correlation between LAB biodegradation and wastewater treatment efficiency and source of contamination in this paragraph. These concepts have been taken too much for granted as well as the meaning of the ratios.

Low levels with LABs degradation were found in the surface sediments of the study environment at SMe3 in Pulau Merambong while high levels were found at SPD1 and SPD2 in Port Dickson.

Add table 2 at the end of the sentence.

Low levels with LABs degradation were found in the surface sediments of the study environment at SMe3 in Pulau Merambong while high levels were found at SPD1 and SPD2 in Port Dickson. The Port Dickson had the highest LAB biodegradation at 64%, while Pulau Merambong had the lowest at 33%. This is explained by the fact that Port Dickson has a higher level of selective biodegradation of LAB species than Pulau Merambong, which has weak degradation and inadequately treated wastewater in some stations.

In my opinion this paragraph is quite redundant, please try to make it more fluid.

This is explained by the fact that Port Dickson has more efficient STPs (Fig.6)

Figure 6?

Pulau Merambong has greater L/S and C13/C12 levels than Port Dickson stations.

Add Figure 6 at the end of the sentence.

Duncan grouping (Table), however there is a difference between C13/C12 and I/E, L/S ratios. These could be attributable to the ratios' sensitivity.

Insert the number of the Table.

How-ever, compared to those found in the Selangor river sediments, the examined area had a greater concentration of C13-LABs, and higher levels of all three markers (0.6, 2.3, and 2.1; Masood et al., 2015)

Please, explain better the “three markers”

The difference in LABs between the the Port Dickson coast and Pulau Merambong

LABs concentration

The SC- homologs, which are less common than LC- ones, are found in the lowest record at the third station of Pulau Merambong

The authors had not used the acronym SC and LC until now, please report in the text.

Fig 2. it is superfluous to report ToC 2 times with 2 different units of measure, eliminate one

Author Response

Reviewer 1

The manuscript evaluates the occurrence and Assessment of Organic Pollutants Residues in the Aquatic Environment of Malesia, in Port Dickson coast and Pulau Merambong which very different level of anthropogenic impact. Even if, the organic chemistry is not my expertise field and some concepts are not easy for me, the article reads well and is well structured, and the results are interesting. However, some revisions are necessary before the publication.

Response:  The authors thank very much the reviewer for the appreciation of the manuscript, which ‎was greatly improved following the reviewer’ comments and suggestions.‎

Introduction

1-  “Due to insufficient sulfonation, LABs are released into the marine environment in a large amount of no treated household and industrial wastes. Therefore, LABs are utilised as anthropogenic pollution indicators because of their enduring and high affinities to industrial effluents”

In the first sentence the authors referred both household and industrial wastes, in the second just to industrial effluent. Is it not incoherent?.‎

Response: The authors thank very much the reviewer for his comments, which was greatly improved the manuscript. As commented, the first and second sentences were restructured and coherent as such " …….because of their enduring ‎and high affinities to household and industrial effluents ".

Page 2   line 72

  • “Due to less growth on Malaysia's eastern coast, the majority of the country's pollution originates in urban and industrial areas”

Please clarify, in which areas is the greater contamination? Industrial and urban areas, but where?

Response: The authors are thankful the reviewer for such a valuable suggestion. As commented, the mentioned sentence was removed since the studied areas were in western coast of Malaysia not in the eastern coast

Page 2 line 87-88

‎3- The researched location underwent measurement of LAB distribution, concentration, and levels of degradation as well as improving the efficiency of the existing STPs.

What are STPs? It is the first time this acronym is reported I the text.

Response: The authors are thankful for such a good comment. STPs was reported in full text as such ‘’ sewage treatment plants’’.

Page 3 line 103

Methods and Materials

  • QUALITY ASSURANCE AND CONTROL

To further avoid any possible sources of cross pollution with the analysis steps, there was a blank sample (four samples per batch) that contained all the standards (SS, IS, and NS) that were present in a regular sample.

NS was not described.

Response: The authors are thankful for such a good comment. NS was known well in the main MS

Page 4 lines 152

  • TOTAL ORGANIC CARBON METHOD (TOC)

In order to completely moisten each weighted dried sediment sample (1-2 g) with 1M HCl, inorganic carbon was removed.

I do not understand, please, rewrite this sentence.

Response: The authors are thankful for such a good comment. The mentioned sentence was rewriting clearly as such ‘’ In order to remove the inorganic carbon, an each weighted dried sediment sample (1-2 ‎g) was completely moisten with 1M HCl.’’.

Page 4 lines 168-169

Results and discussion

  • COMPOSITION, DISTRIBUTION AND CONCENTRATION of LAB

Total number of LABs C10–C14 found in samples from all sites under study (Fig.2).

Maybe “Total number of LABs C10–C14 found in samples from all sites under study were reported in Figure 2”

Response: The authors are thankful the reviewer for such a valuable suggestion. As commented, the mentioned sentence was adjusted to your suggestion as such “Total number of LABs C10–C14 found in samples from all sites under study were reported in Figure 2”

Page 4 lines 176-177

  • Lengthy-chains LAB (C13 and C14), as opposed to other LAB homologs (C10 and C11), tend to prevail in marine environments, demonstrating that LABs have long lateral transit in this area.

What the authors exactly mean with “demonstrating that LABs have long lateral transit in this area”. Please, explain better.

Response: The authors are thankful for such a good comment. The mentioned sentence was clearly explained as such ‘’ The abundant of lengthy-chains LAB (C13 and C14), compared to short LAB homologs (C10 and C11), tend to prevail in marine environments, demonstrating that LABs have long lateral transport in this area’’.

Page 4 lines 181-183

  • Table 2 demonstrates the presence of LABs C10-14 in surface sediments varied between 67.4 in Pulau Merambong's SMe3 to 255.8 ng g−1dw in Port Dickson's SPD1 (Fig.4). SMe3 in Pulau Merambong had the lowest concentration of LABs, whereas first station (SPD1) of Port Dickson had the highest.

These two sentences are redundant, please merge them.

Response: The authors are thankful for such a good comment. As suggested, the above sentences were merged as such ‘’ Table 2 demonstrates the presence of LABs C10-14 in surface sediments varied between 67.4 in Pulau Merambong's SMe3 to 255.8 ng g−1dw in Port Dickson's SPD1 (Fig.4), showing that  SMe3 in Pulau Merambong had the lowest concentration of LABs, whereas first station (SPD1) of Port Dickson had the highest’’.

Page 5 lines 197- 200

  • TOC EVALUATION

All sediment samples had TOC assessed, and several locations, including Port Dick-son and Pulau Merambong had a relationship between LABs and TOC

What the authors mean with several locations?

Response: The authors are thankful the reviewer for such a valuable suggestion. Several locations mean most of locations in studied areas, the mentioned sentence was adjusted as such “All sediment samples had TOC assessed, and most stations in Port Dickson and Pulau Merambong had a relationship between LABs and TOC”

Page 5 lines 227-228

  • SMe2 of Pulau Merambong had the lowest concentration of TOC, whereas SMe3 in the same island had the highest. The Pulau Merambong's SMe2 had TOC with 6.5 mg g-1 and SMe3 in the same island had TOC with 35.1 mg/g..

Please, merge these two sentences.

Response: As suggested, the mentioned sentences were merged as such ‘’ The Pulau Merambong's SMe2 had TOC with 6.5 mg g-1 and SMe3 in the same island had TOC with 35.1 mg g-1, implying that SMe2 of Pulau Merambong had the lowest concentration of TOC, whereas SMe3 in the same island had the highest’’.

Page 5 lines 236-238

  • ASSESSMENT OF LAB DEGRADATION AND EFFLUENTS TREATMENTS EFFICIENCY

please, try to explain better the correlation between LAB biodegradation and wastewater treatment efficiency and source of contamination in this paragraph. These concepts have been taken too much for granted as well as the meaning of the ratios.

Response: As suggested, more explain for the correlation between LAB biodegradation and wastewater treatment efficiency and source of contamination were added,

Page 6 lines 246-258

  • Low levels with LABs degradation were found in the surface sediments of the study environment at SMe3 in Pulau Merambong while high levels were found at SPD1 and SPD2 in Port Dickson.

Add table 2 at the end of the sentence.

Response: As suggested, Table 2 was added to the sentence

Page 6 line 260

  • Low levels with LABs degradation were found in the surface sediments of the study environment at SMe3 in Pulau Merambong while high levels were found at SPD1 and SPD2 in Port Dickson. The Port Dickson had the highest LAB biodegradation at 64%, while Pulau Merambong had the lowest at 33%. This is explained by the fact that Port Dickson has a higher level of selective biodegradation of LAB species than Pulau Merambong, which has weak degradation and inadequately treated wastewater in some stations.

In my opinion this paragraph is quite redundant, please try to make it more fluid.

Response: As suggested, the mentioned sentences were amended fluidly as such’’ The LABs degradation were found with Low levels in the surface sediments at SMe3 in Pulau Merambong while high levels were found at SPD1 and SPD2 in Port Dickson (Table 2). The study results showing that the Port Dickson had the highest LAB biodegradation at 64%, while Pulau Merambong had the lowest at 33%, indicating that Port Dickson has a higher level of selective biodegradation of LAB species than Pulau Merambong, which has weak degradation and inadequately treated wastewater in some stations’’.

Page 6 lines 258-264

  • This is explained by the fact that Port Dickson has more efficient STPs (Fig.6)

Figure 6?

Response: As suggested, Fig.6 changed to Figure 6

Page 5 line 245

  • Pulau Merambong has greater L/S and C13/C12 levels than Port Dickson stations.

Add Figure 6 at the end of the sentence

Response: As suggested, Figure 6 was added to end of the sentence

Page 6 line 269

  • Duncan grouping (Table), however there is a difference between C13/C12 and I/E, L/S ratios. These could be attributable to the ratios' sensitivity.

Insert the number of the Table.

Response: The authors are thankful the reviewer for such a valuable suggestion. The number of the Table was inserted at the end of the sentence

Page 6 line 285

  • How-ever, compared to those found in the Selangor river sediments, the examined area had a greater concentration of C13-LABs, and higher levels of all three markers (0.6, 2.3, and 2.1; Masood et al., 2015)

Please, explain better the “three markers”

Response: The authors are thankful for such a good comment. The three markers (I/E, L/S and C13/C12)  of this study were compared clearly with those markers in Selangor river study as such ‘’  When compared these markers (I/E; 4.1, L/S;2.0 and C13/C12; 14.3) to those found in the Selangor river sediments, this study had a greater concentration of C13-LABs, and higher levels of all three markers (0.6, 2.3, and 2.1; Masood et al., 2015)

Page 6 lines 295-298

  • The difference in LABs between the Port Dickson coast and Pulau Merambong

LABs concentration

Response: As commented, this sentence was amended as such’’ The levels of LABs between the Port Dickson coast and Pulau Merambong stations’’

Page 7 line 308

  • The SC- homologs, which are less common than LC- ones, are found in the lowest record at the third station of Pulau Merambong

The authors had not used the acronym SC and LC until now, please report in the text.

Response: The authors are thankful for such a good comment.  The acronym SC and LC were reported in result and discussion in particular in section ‘’ COMPOSITION, DISTRIBUTION AND CONCENTRATION’’

  • Fig 2. it is superfluous to report ToC 2 times with 2 different units of measure, eliminate one

Response: The authors are thankful for such a good comment. As suggested, one unit of TOC measurement was eliminated

Table 2

Reviewer 2 Report

The manuscript investigates linear alkyl benzene as an organic pollutant in the coastal sediments of Malaysia. The authors used GC-MS as the major analytical method to identify LABs in the sediments. Long to short and internal to external chains were applied to study the degradation rate of LABs. They concluded that the wastewater systems need improvement and LAB molecular markers are quite effective in tracing contaminants. 

The study is quite interesting, but I still have some challenging doubts regarding the work.

1. It is suggested to add a sample formula for linear alkyl benzene in the manuscript to help the readers. 

2. I am not satisfied with the resolution of Figure 2. 

3.  There are some subscript and superscript problems in the x-axis and y-axis of some figures.

4. The TOC calculations seem to be a determining factor in your study on the basis of Figure 5. However, in the research you cited it wasn't. please discuss.

5. What other analytical methods have been used by researchers in similar studies conducted? You can add a paragraph in introduction part.  

Overall, the manuscript needs minor revision from my perspective.

Author Response

Reviewer 2

The manuscript investigates linear alkyl benzene as an organic pollutant in the coastal sediments of Malaysia. The authors used GC-MS as the major analytical method to identify LABs in the sediments. Long to short and internal to external chains were applied to study the degradation rate of LABs. They concluded that the wastewater systems need improvement and LAB molecular markers are quite effective in tracing contaminants.

The study is quite interesting, but I still have some challenging doubts regarding the work.

Response: ‎The authors thank very much the reviewer for giving us the opportunity to improve our manuscript, which ‎was greatly improved following the reviewer’ comments and suggestions.‎

  • It is suggested to add a sample formula for linear alkyl benzene in the manuscript to help the readers.

Response: The authors thank very much the reviewer for such a valuable comment. As suggested, the formula for linear alkyl benzene was added in intrudation section (C6H5-CnH2n+l, n= 10-14)

Page 2   Line 65

  • I am not satisfied with the resolution of Figure 2.

Response: The authors thank very much the reviewer for such good observation. As suggested, the figure 2 replaces with high quality resolution

Figure 2

3- There are some subscript and superscript problems in the x-axis and y-axis of some figures.

Response: The authors are thankful for such a good comment. As suggested, the subscripts in the x-axis and y-axis in the figures were checked properly

  • The TOC calculations seem to be a determining factor in your study on the basis of Figure 5. However, in the research you cited it wasn't. please discuss.

Response: The authors are thankful for such a good comment. As suggested, the TOC was cited as a determining factor for LABs distribution in the study area based on Figure 5

Page 5 lines 242-243

  • What other analytical methods have been used by researchers in similar studies conducted? You can add a paragraph in introduction part.

Response: Thank you for your valuable comment. Based on your comment, some information of other studies was added in introduction part,  

Pages 1-2 Lines 41-63